# Tacrolimus (FK506) Exhibits Fungicidal Effects against *Candida parapsilosis* Sensu Stricto via Inducing Apoptosis

**DOI:** 10.3390/jof9070778

**Published:** 2023-07-24

**Authors:** Otomi Cho, Shintaro Takada, Takahiro Odaka, Satoshi Futamura, Sanae Kurakado, Takashi Sugita

**Affiliations:** Department of Microbiology, Meiji Pharmaceutical University, Noshio, Kiyose 204-8588, Japan

**Keywords:** calcineurin inhibitors, apoptosis, FK506, *Candida parapsilosis*

## Abstract

Tacrolimus (FK506), an immunosuppressant and calcineurin inhibitor, has fungicidal effects. However, its fungicidal effect is thought to be limited to basidiomycetes, such as *Cryptococcus* and *Malassezia*, and not to ascomycetes. FK506 had no fungicidal effect on *Candida albicans*, *C. auris*, *C. glabrata*, *C. guilliermondii*, *C. kefyr*, *C. krusei*, and *C. tropicalis* (>8 µg/mL); however, *C. parapsilosis* was susceptible to it at low concentrations of 0.125–0.5 µg/mL. *C. metapsilosis* and *C. orthopsils*, previously classified as *C. parapsilosis*, are molecularly and phylogenetically closely related to *C. parapsilosis*, but neither species was sensitive to FK506. FK506 increased the mitochondrial reactive oxygen species production and cytoplasmic and mitochondrial calcium concentration and activated metacaspases, nuclear condensation, and DNA fragmentation, suggesting that it induced mitochondria-mediated apoptosis in *C. parapsilosis*. Elucidating why FK506 exhibits fungicidal activity only against *C. parapsilosis* will provide new information for developing novel antifungal drugs.

## 1. Introduction

Calcineurin is a conserved Ca^2+^-calmodulin-activated protein phosphatase involved in calcium-dependent signaling and the regulation of several important cellular processes in fungi. Calcineurin phosphatase activity is enhanced by the binding of the activated Ca^2+^-calmodulin complex to a catalytic subunit (calcineurin A; CaN) and a regulatory subunit (calcineurin B; CnB); however, calcineurin activity is inhibited by the binding of tacrolimus (FK506) and cyclosporin A (CsA) to FKBP12 and CypA, respectively [1,2,3]. Both calcineurin inhibitors, FK506 and CsA, have been used to prevent organ transplant rejection (kidney and liver) and graft-versus-host reactions. They exhibit temperature sensitivity and have a fungicidal effect on the pathogenic yeast *Cryptococcus neoformans.* Specifically, they display this effect at 37 °C but not at 27 °C [4,5]. Mortality from cryptococcosis in organ transplant patients treated with FK506 was significantly lower than that in patients treated with the immunosuppressive non-calcineurin inhibitor azathioprine, suggesting that calcineurin inhibitors have potential antifungal activity [6]. For filamentous fungi, FK506 demonstrated inhibitory effects on the growth of *Aspergillus fumitatus* in vitro. This was further supported by experiments using a mouse infection model, where the FK506-treated group showed significant survival extension compared with that of the control group [7].

Candidiasis is an opportunistic infection caused by *Candida* species, which colonize the human digestive tract, skin, or oral cavity, primarily affecting immunocompromised hosts. The causative agents are diverse, with the majority comprising *Candida albicans*, *C. glabrata*, and *C. parapsilosis*. As the number of immunocompromised hosts increases, the global incidence of bloodstream fungal infections is also rising, leading to candidiasis being recognized as one of the two major fungal diseases, along with aspergillosis [8,9]. *Candida* species have the capacity to form biofilms on medical devices such as catheters and artificial joints, thus rendering treatment of these biofilm infections complex [10]. Notably, *C. parapsilosis* is prevalent on the skin, facilitating catheter transfer [11]. FK506 exhibits growth-inhibitory effects against *Cryptococcus neoformans* and *Aspergillus fumigatus* [6,7]. However, in the case of *C. albicans*, *C. glabrata*, and *C. krusei*, these effects are only observed when FK506 is combined with azole antifungal drugs and not when FK506 is used alone [12,13,14]. Consequently, it has been presumed that FK506 does not exert a direct growth-inhibitory effect on ascomycetous yeasts.

During our investigation into the growth-inhibitory impact of FK506 on ascomycetous yeasts, we discovered that FK506 exerts a direct effect on *C. parapsilosis* when treated alone, an effect mediated via apoptosis.

## 2. Materials and Methods

### 2.1. Strains and Drug Susceptibility Testing

The susceptibilities to tacrolimus (FK506; FUJIFILM Wako Pure Chemical Corporation, Osaka, Japan) and CsA (FUJIFILM Wako Pure Chemical Corporation) of 21 strains of *C. parapsilosis* and 46 other pathogenic ascomycetous yeasts, listed in Table 1, were tested using the broth microdilution method in accordance with CLSI M27-A3 [15]. Both FK506 and CsA were dissolved in dimethyl sulfoxide (DMSO). Test microtiter plates with 96 U wells were incubated for 2 days at 27 °C and 37 °C with drug concentrations of 0.03–8 μg/mL. The minimum inhibitory concentration was defined as the lowest concentration that completely inhibited growth. Of the 67 strains, 2 (*C. albicans* SC5314 and *C. parapsilosis* J2-104) were used to elucidate the mechanism of the fungicidal action of FK506.

### 2.2. Yeast Cell Survival Assay Using Fluorescence Microscopy

The effect on survival was determined using a two-color fluorescent probe, FUN1 Cell Stain (Life Technologies, Eugene, OR, USA), and fungal surface labeling with Calcofluor White Stain (CFW, 1 mg/mL; Sigma-Aldrich, St. Louis, MO, USA) according to the manufacturer’s instructions. Briefly, 1 × 10^5^ cells/mL of *C. parapsilosis* were incubated with FK506 at a concentration of 0.5 μg/mL. After incubation for 0, 3, and 6 h at 37 °C, the cells were centrifuged at 12,000 rpm for 1 min and washed once with GP solution (2% glucose in PBS). Thereafter, the GP solution was replaced with a GP solution containing 5 μM FUN1 and 2 μL CFW. After incubation for 30 min at room temperature, the cells were observed under a fluorescence microscope (BX61; Olympus Corporation, Tokyo, Japan) with a DP70 camera (Olympus Corporation). Staining with FUN1 (excitation = 488 nm and emission = 530 nm) and CFW (excitation = 365 nm and emission = 440 nm) was performed using WIB and WU filter cubes (Olympus Corporation).

### 2.3. Measurement of Mitochondrial Reactive Oxygen Species

Mitochondrial reactive oxygen species (ROS) accumulation was assessed using a MitoSOX Red Mitochondrial Superoxide Indicator (Life Technologies) according to the manufacturer’s instructions. Briefly, 1 × 10^6^ cells/mL of *C. albicans* and *C. parapsilosis* cells were treated with FK506 (0.25, 0.5, and 2 μg/mL) at 37 °C for 3 h. Subsequently, the cells were harvested via centrifugation at 12,000 rpm and suspended in PBS containing 5 μM MitoSOX Red at 37 °C for 15 min. MitoSOX Red fluorescence (excitation = 510 nm, emission = 580 nm) was observed using a WIG filter cube (Olympus Corporation).

### 2.4. Measurement of Ca^2+^ Levels in Cytosol and Mitochondria

Fluo-8^TM^/AM (AAT Bioquest, Inc., Sunnyvale, CA, USA) and Rhod-2/AM (Dojindo, Kumamoto, Japan) were used to analyze cytosolic and mitochondrial Ca^2+^ levels, respectively [16,17]. Briefly, 1 × 10^6^ cells/mL of *C. parapsilosis* cells were incubated with FK506 at concentrations of 0, 0.5, and 2 μg/mL, as described above, at 37 °C for 3 h. Next, the cells were washed twice with Krebs buffer (132 mM NaCl, 4 mM KCl, 1.4 mM MgCl_2_, 6 mM glucose, 10 mM HEPES, 10 mM NaHCO_3_, and 1 mM CaCl_2_; pH 7.2; Sigma-Aldrich) containing 0.1% Pluronic F-127 (Molecular Probes, Eugene, OR, USA) and 1% bovine serum albumin (FUJIFILM Wako Pure Chemical Corporation). Thereafter, the suspensions were incubated with 5 mM Fluo-8^TM^/AM or 10 mM Rhod-2/AM at 27 °C for 30 min and washed thrice with calcium-free Krebs buffer. The fluorescence intensities of Fluo-8^TM^/AM (excitation = 485 nm and emission = 538 nm) and Rhod-2/AM (excitation = 544 nm and emission = 590 nm) were measured using a Shimadzu RF-5301PC Spectrofluorophotometer (Shimadzu, Kyoto, Japan). The suspensions were then observed under a fluorescence microscope.

### 2.5. Detection of Metacaspase Activity

Activated metacaspases in *C. parapsilosis* cells were measured using a CaspACE FITC-VAD-FMK in situ marker (Promega, Madison, WI, USA) [18]. Briefly, 1 × 10^6^ cells/mL of the microorganisms were treated with 0 and 0.5 μg/mL of FK506 for 6 h at 37 °C. Afterward, the cell suspensions were washed with PBS and stained with 10 mM CaspACE FITC-VAD-FMK for 30 min at 37 °C. Fluorescence was observed using a WIB filter cube (excitation = 488 nm and emission = 520 nm; Olympus Corporation).

### 2.6. DNA and Nuclei Damages Assay

Nuclear condensation and fragmentation were analyzed using DAPI staining (Dojindo) and terminal deoxynucleotidyl transferase dUTP nick-end labeling (TUNEL) assays [19]. *C. parapsilosis* cells (1 × 10^6^ cells/mL) were incubated with 0 and 0.5 μg/mL of FK506 for 6 h at 37 °C. For DAPI staining, the cells were washed twice with PBS and treated with DAPI solution (Dojindo) for 15 min in the dark. For TUNEL staining, cells were washed twice with PBS, fixed in 4% paraformaldehyde, permeabilized on ice for 2 min, and washed again with PBS. The DNA ends were labeled with an In Situ Cell Death Detection Kit (Sigma-Aldrich) for 1 h at 37 °C. The cells were then harvested and examined under a fluorescence microscope. DAPI (excitation = 365 nm and emission = 460 nm) and TUNEL staining (excitation = 488 nm and emission = 535 nm) were performed using WU and WIB filter cubes (Olympus Corporation).

### 2.7. Statistical Analysis

Data are expressed as mean ± standard deviation (*n* = 3) and were analyzed statistically using Student’s *t*-test. *p* < 0.05 was considered statistically significant. Analyses were conducted using R version 3.6.3 (R Foundation for Statistical Computing, Vienna, Austria).

## 3. Results

### 3.1. Antifungal Activity of FK506 and CsA

Although all 15 strains of *C. parapsilosis* sensu stricto showed susceptibility to FK506 at 0.125–0.5 µg/mL at both 27 °C and 37 °C, *C. metapsilosis* and *C. orthopsilosis*, which were previously classified as *C. parapsilosis*, showed resistance (>8 μg/mL) to FK506 (Table 1). All other pathogenic *Candida* species also showed resistance (>8 μg/mL) to FK506. Regarding CsA sensitivity, *C. parapsilosis* and all other *Candida* species exhibited no resistance at 27 °C and 37 °C.
jof-09-00778-t001_Table 1Table 1Drug susceptibilities of FK506 and CsA against *Candida parapsilosis* and others pathogenic *Candida* species.SpeciesNumber of StrainsFK506 (µg/mL)CsA (µg/mL)27 °C37 °C27 °C37 °C*Candida parapsilosis* sensu stricto150.125–0.50.125–0.5>8>8*Candida metapsilosis*4>8>8>8>8*Candida orthopsilosis*2>8>8>8>8*Candida albicans*6>8>8>8>8*Candida auris*2>8>8>8>8*Candida dubliniensis*5>8>8>8>8*Candida glabrata* (=*Nakaseomyces glabratus*)7>8>8>8>8*Candida guilliermondii* (=*Meyerozyma guilliermondii*)6>8>8>8>8*Candida kefyr* (=*Kluyveromyces marxianus*)4>8>8>8>8*Candida krusei* (=*Pichia kudriavzevii*)6>8>8>8>8*Candida lusitaniae* (=*Clavispora lusitaniae*)1>8>8>8>8*Candida tropicalis*7>8>8>8>8*Candida viswanathii*1>8>8>8>8*Candida norvegensis* (=*Pichia norvegensis*)1>8>8>8>8While all yeast species names are listed under the genus *Candida*, names corresponding to the most recent taxonomy are provided in parentheses.


### 3.2. Fluorescence Staining for Cell Viability

The viability of *C. parapsilosis* cells was evaluated via FUN1staining. The cell walls were stained with CFW to clearly observe the cell morphology, and the FUN1- and CFW-stained cell images were merged (Figure 1a). There were no differences observed in the cellular morphology between *C. parapsilosis* FK506-treated and untreated cells. *C. parapsilosis* cells treated with FK506 (6 h) exhibited decreased metabolism compared with that of control cells. The viability of FK506-untreated cells did not change at 3 and 6 h, whereas that of FK506-treated cells decreased in a time-dependent manner to 57.8 ± 23.7% at 3 h and 42.5 ± 14.4% at 6 h (Figure 1b).

### 3.3. Mitochondrial ROS Production in FK506-Treated Cells

Accumulated ROS causes oxidative damage to essential biomolecules [20]. Since MitoSOX Red is a specific ROS indicator in the mitochondria, red fluorescence was observed in FK506-treated cells (Figure 2a). FK506 affected the mitochondria of *C. parapsilosis* but not those of *C. albicans*. Treatment with FK506 for 3 h increased mitochondrial ROS production in *C. parapsilosis* by 75.5% at 0.25 μg/mL and 81.0% at 2 μg/mL (Figure 2b). However, *C. albicans* barely produced ROS upon treatment with FK506.

### 3.4. Measurement of Ca^2+^ Levels

Ca^2+^ is an important messenger in organisms because it is involved in a series of biological processes, including cell growth, proliferation, apoptosis, and mating morphogenesis [21]. In the presence of FK506, *C. parapsilosis* cells stained green or red with Fluo-8/AM or Rhod-2/AM staining were observed, respectively (Figure 3a,b).

The cells were treated with 0.5 or 4 g/mL FK506 for 3 h; cytosolic Ca^2+^ increased 3.7- and 4.3-fold, respectively, and mitochondrial Ca^2+^ levels increased 6.1- and 6.3-fold, respectively (Figure 3a,c).

These results indicate that FK506 induced the elevation of cytosolic and mitochondrial Ca^2+^ levels, indicating the involvement of calcium signaling in fungal cell death.

### 3.5. Metacaspase Activity

The FITC-labeled metacaspase inhibitor VAD-FMK irreversibly binds to activated metacaspases in apoptotic cells [22]. The number of *C. parapsilosis* cells stained with CaspACE FITC-VAD-FMK increased in an FK506 concentration-dependent manner (Figure 4). These results demonstrate that FK506 activates the metacaspase-dependent pathway in *C. parapsilosis* cells.

### 3.6. Nuclear Fragmentation and DNA Damage

DNA fragmentation and nuclear condensation and fragmentation are characteristics of late apoptosis. We confirmed DNA fragmentation and condensation in *C. parapsilosis* cells using DAPI and TUNEL assays [19,23]. No TUNEL-positive nuclei (green fluorescence) were observed in FK506-non-treated cells, whereas the number of TUNEL-positive nuclei showed an increasingly concentrated fluorescence intensity in single cells with increasing FK506 concentration, which is indicative of DNA damage induced by FK506 in *C. parapsilosis* (Figure 5). These findings were corroborated via DAPI staining. Similarly, *C. parapsilosis* cells exposed to FK506 showed a DAPI-positive phenotype and chromatin condensation, indicating nuclear condensation and DNA fragmentation. Based on our results, FK506 induced changes in the structure and content of nuclear DNA in *C. parapsilosis*.

## 4. Discussion

FK506, a calcineurin inhibitor, has a direct fungicidal effect against the ascomycetous yeast *C. parapsilosis* among pathogenic *Candida* species. We hypothesized that FK506 inhibits the growth of *C. parapsilosis* by inducing apoptosis. In the presence of FK506, the following phenomena were observed: (1) increased mitochondrial ROS production, (2) increased cytosolic and mitochondrial calcium concentrations, (3) metacaspase activation, (4) nuclear condensation, and (5) DNA fragmentation. These phenomena suggest that FK506 induces mitochondria-mediated apoptosis in *C. parapsilosis*.

In fungi, apoptosis is initiated by an increase in the intracellular calcium ion concentration. When cells are stressed, extracellular calcium ions enter the cytoplasm and are transported to the mitochondria. Mitochondria are depolarized, and permeability transition pore formation is triggered by calcium overload [24]. In the present study, FK506 treatment increased the number of Fluo-8/AM- and Rhod-2/AM-positive cells, suggesting that FK506 treatment induces cell stress, causing calcium influx into the cytoplasm and inducing apoptosis. ROS production occurs in the early phase of apoptosis, when the intracellular calcium ion concentration increases and mitochondrial load is applied [25]. Mitochondria are important for intracellular ROS production, and excessive ROS production causes oxidative stress in cells [26]. Intracellular ROS accumulation causes cell death in yeast [25,27]. MitoSOX staining showed that FK506 treatment resulted in intracellular ROS production and accumulation in *C. parapsilosis*; however, these phenomena were not observed in *C. albicans*. Caspase activation plays a significant role in apoptosis [28]. In mammals, cytochrome c-activated caspase activator (Apaf1) activates caspase-9, which acts as an apoptosis initiator, and caspase-3, -6, and -7, which degrade intracellular proteins. However, its ortholog metacaspase is conserved in fungi [28,29]. Since intracellular ROS are also involved in metacaspase activation, metacaspase activation is promoted by an increase in intracellular ROS production [30]. The caspase-ACE FITC-VAD-FMK in situ marker used in this study emitted green fluorescence upon binding to the active site of the activated metacaspase. Positive cells were confirmed 6 h after FK506 treatment, suggesting that FK506 induced apoptosis in *C. parapsilosis* via metacaspases. DNA damage and nuclear fragmentation during the late phase of apoptosis are typical morphological features of apoptotic cells [31]. Activation of caspases leads to the proteolytic degradation of nuclear proteins, damaging chromatin and causing DNA damage and chromatin condensation [32].

FK506 has a fungicidal effect only on *C. parapsilosis* sensu stricto and shows no effect on *C. metapsilosis* or *C. orthopsilosis*, which were formally classified as *C. parapsilosis* [33]. It is of evolutionary interest that the nucleotide sequences of the ITS regions of the rRNA genes of the three species have a similarity of 98% or more, and although they are phylogenetically closely related, only *C. parapsilosis* shows sensitivity to FK506.

In conclusion, our study demonstrated that FK506 induces apoptosis, specifically in *C. parapsilosis*, providing a basis for its fungicidal effect. This effect involves mitochondrial ROS production, increased calcium concentration, metacaspase activation, and DNA fragmentation. This discovery could pave the way for new treatments against *C. parapsilosis*.

## Figures and Tables

**Figure 1 jof-09-00778-f001:**
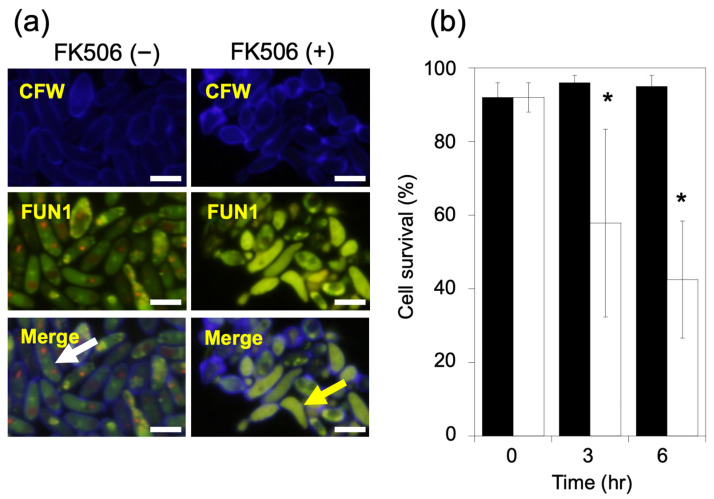
Viability of *C. parapsilosis* strain cells using FUN1 and CFW. (**a**). To visualize fungal components, *Candida parapsilosis* viability was imaged via microscopy using calcofluor white (CFW) and FUN1. Green fluorescence accumulated throughout the cytoplasm, and red particles transferred and concentrated in the cytoplasmic vacuoles, indicating metabolic activity. Scale bars, 10 μm. White and yellow arrows indicate viable and dead cells, respectively. (**b**). *C. parapsilosis* was incubated with or without 0.5 μg/mL of FK506 at 37 °C for 3 or 6 h. Viability was calculated as the ratio of the number of cells showing red fluorescence (stained with FUN1) to the total number of cells (stained blue with CFW). Statistical analysis to compare with the untreated controls was conducted. Data are represented as mean ± standard deviation. Two-hundred-and-fifty cells were measured per treatment. Experiments were conducted in triplicate. Closed square, FK506 treated; open square, FK506 untreated; * *p* < 0.05, compared with untreated controls.

**Figure 2 jof-09-00778-f002:**
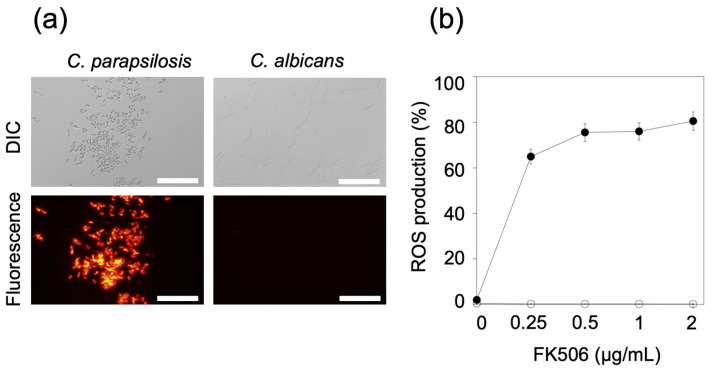
Measurement of mitochondrial ROS production in FK506-treated cells. (**a**) Mitochondrial ROS production was observed using MitoSOX Red. Following treatment with FK506 (0.5 μg/mL) for 3 h, the cells were imaged using a microscope. Scale bars, 100 μm. (**b**) Mitochondrial ROS production ratio of FK506-treated *Candida parapsilosis* and *C. albicans* cells (0, 0.25, 0.5, 1, and 2 μg/mL) was measured at 37 °C for 3 h. The ratio was calculated as the number of cells showing red fluorescence (stained with MitoSOX Red) to the total number of cells (approximately 300 cells). Data are presented as mean ± standard deviation. Experiments were conducted in triplicate. Closed circle, *C. parapsilosis* cells; open circle, *C. albicans* cells.

**Figure 3 jof-09-00778-f003:**
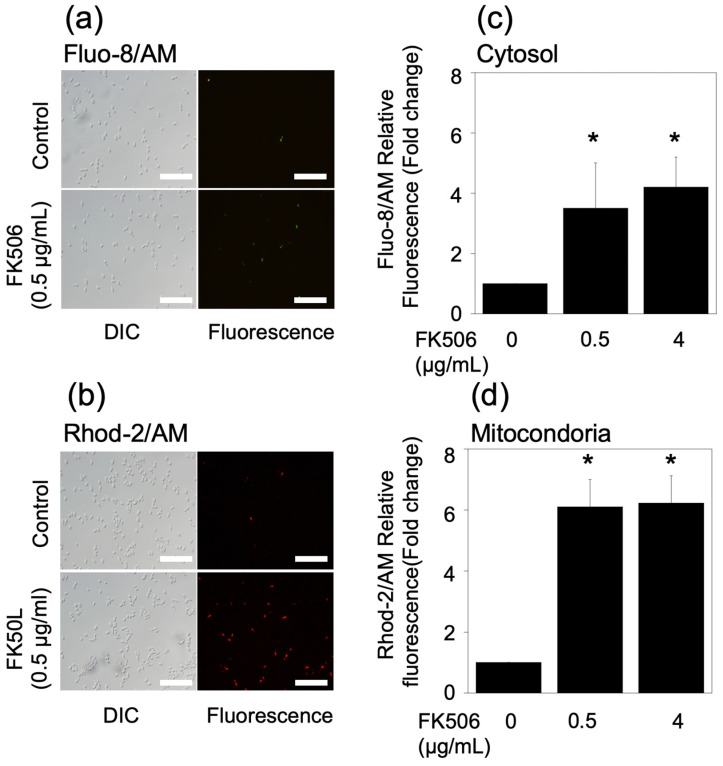
Measurement of cytosolic and mitochondrial Ca^2+^ levels. (**a**,**b**) Fluorescence microscopy image in cytosolic (green fluorescence) and mitochondrial (red fluorescence) Ca^2+^ levels at 0 and 0.5 μg/mL FK506 at 37 °C for 3 h, respectively. Scale bars, 100 μm. (**c**,**d**) Cytosolic and (**c**) and mitochondrial (**d**) Ca^2+^ levels at 0, 0.5, and 4 μg/mL FK506 at 37 °C for 3 h, respectively. Data represent mean ± standard deviation. Experiments were conducted in triplicate. * *p* < 0.05 compared with untreated controls.

**Figure 4 jof-09-00778-f004:**
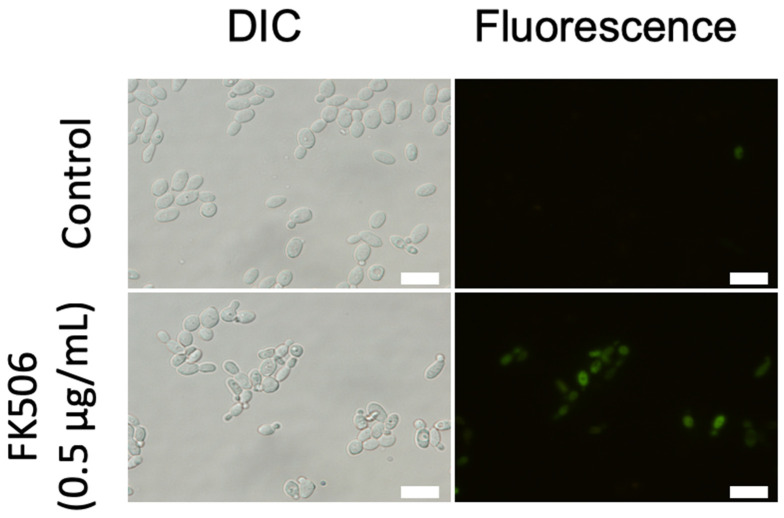
Metacaspase detection using CaspACE FITC-VAD-FMK. After *Candida parapsilosis* was incubated with 0.5 μg/mL of FK506 at 37 °C for 6 h, the cells were labeled with the CaspACE FITC-VAD-FMK reagent and examined using fluorescence microscopy. Metacaspase activity was indicated using green fluorescence staining. Scale bars, 10 μm.

**Figure 5 jof-09-00778-f005:**
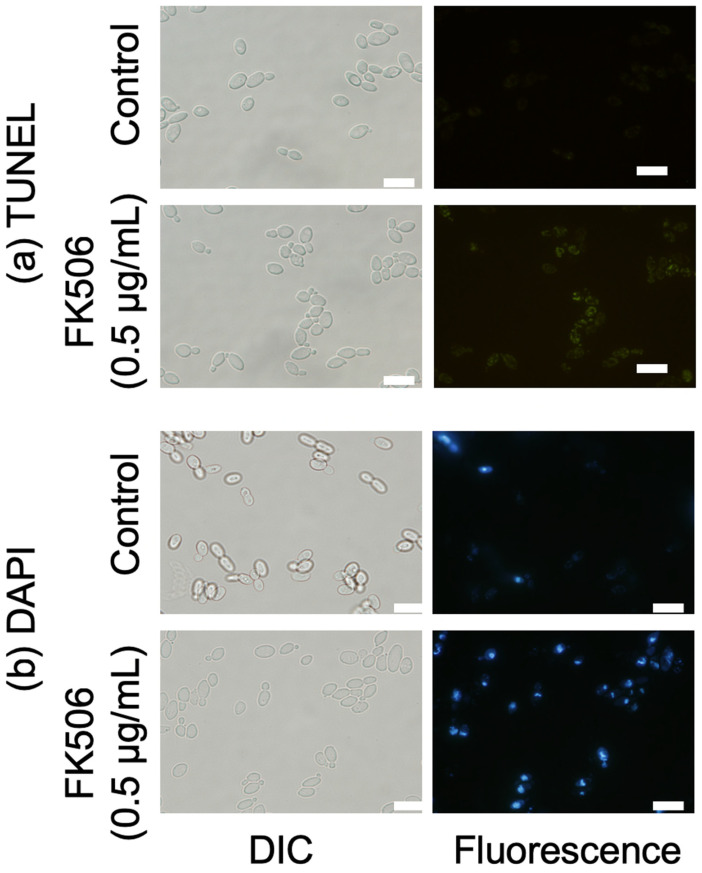
Detection of late apoptotic features using fluorescence microscopy. The cells were labeled after *Candida parapsilosis* was incubated with 0.5 μg/mL FK506 at 37 °C for 6 h. (**a**) DNA fragmentation observed using the TUNEL assay (green fluorescence). (**b**) Nuclear condensation and fragmentation visualized using DAPI (blue fluorescence). Scale bars, 10 μm.

## Data Availability

Not applicable.

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
