# Peer review of "Tacrolimus (FK506) Exhibits Fungicidal Effects against *Candida parapsilosis* Sensu Stricto via Inducing Apoptosis"

_jof, 2023, doi:10.3390/jof9070778_

Round 1
Reviewer 1 Report
Althoug it is interesting that C. parapsilosis responds differentially to Tacrolimus, the manuscript would improve if the authors assay mutants or do any molecular determination to increase the knowledge on the molecular mechanisms of action of Tacrolimus.
Author Response
Dear Reviewers,
Thank you very much for your suggestions and comments; we have modified the manuscript accordingly. The modified sentences have been highlighted in yellow in the manuscript.
Reviewer 1
Although it is interesting that C. parapsilosis responds differentially to Tacrolimus, the manuscript would improve if the authors assay mutants or do any molecular determination to increase the knowledge on the molecular mechanisms of action of Tacrolimus.
Response: Thank you very much for your suggestion.
As you pointed out, if we can elucidate the direct antifungal effects of FK506 on C. parapsilosis at the genetic level, we believe it will contribute to future drug discovery research. This study is the first to reveal that among the ascomycetous yeasts, only C. parapsilosis exerts its antifungal effects through apoptosis. Going forward, we plan to conduct research at the genetic level.
Reviewer 2 Report
The authors tried to characterize how tacrolimus (FK506) fungicidal effect against Candida parapsilosis in growth phenotype. And the effects of genetic profile on A. fumigatus were determined. Their findings are important for a potent antifungal agent of FK506 in treatment of C. parapsilosis. I provided some comments for the authors to consider as outlined below.
General comments:
1) More information about the disease, caused by C. parapsilosis, and about the different between C. parapsilosis and other Candida species, should be provide in the Introduction.
2) The viability of FK506 treated cells were performed at 3 h and 6 h. How about the mitochondria ROS production, cytosolic, and mitochondrial Ca2+ levels, in FK506-treated cells at 6 h? And how about the changes of metacaspase and apoptotic at 3 h?
3) More expression information of these genes or protein, related to these morphologies, in the results should perform.
4) These is repeat results describe in the DISCUSSION. Therefore, the part should be revised. And Line 225-249 should be merged to new paragraphs.
5) It was confirmed that FK506 had a good antifungal effect against C. parapsilosis in vitro. However, more information of FK506 against A. fumigatus in vivo should be investigated.
6) These are no data of Table 1 in the manuscript.
Other comments:
1. Line 67: The reference of the manufacturer’s instructions should be provided.
2. Line 106: All the statistical were performed using which software?
3. Line 122, 141: The data should be shown with mean ± standard deviation.
4. Line 248-249: Reference should be provided.
Author Response
Dear Reviewers,
Thank you very much for your suggestions and comments; we have modified the manuscript accordingly. The modified sentences have been highlighted in yellow in the manuscript.
Reviewer 2
The authors tried to characterize how tacrolimus (FK506) fungicidal effect against Candida parapsilosis in growth phenotype. And the effects of genetic profile on A. fumigatus were determined. Their findings are important for a potent antifungal agent of FK506 in treatment of C. parapsilosis. I provided some comments for the authors to consider as outlined below.
General comments:
- More information about the disease, caused by C. parapsilosis, and about the different between C. parapsilosis and other Candidaspecies, should be provide in the Introduction.
Response: We have added information regarding candidiasis and C. parapsilosis to the Introduction according to your suggestion. (Lines 38–51)
- The viability of FK506 treated cells were performed at 3 h and 6 h. How about the mitochondria ROS production, cytosolic, and mitochondrial Ca2+levels, in FK506-treated cells at 6 h? And how about the changes of metacaspase and apoptotic at 3 h?
Response: The production of mitochondrial ROS and Ca2+ levels can be observed at 3 h; therefore, we did not include data from observations at 6 h. Metacaspase activity and DNA and nucleus damage could not be observed at 3 h; therefore, we used observations at 6 h.
- More expression information of these genes or protein, related to these morphologies, in the results should perform.
Response: We have added this information to the manuscript as suggested.
There were no differences observed in the cellular morphology between C. parapsilosis FK506-treated and untreated cells. (Lines 137-138)
- These is repeat results describe in the DISCUSSION. Therefore, the part should be revised. And Line 225-249 should be merged to new paragraphs.
Response: We have reviewed the entire document again, removed the aforementioned text from the Results section, and organized the information in lines 225 to 249 according to your suggestion.
- It was confirmed that FK506 had a good antifungal effect against C. parapsilosis in vitro. However, more information of FK506 against A. fumigatusin vivoshould be investigated.
Response: This study focuses specifically on pathogenic Candida species and does not discuss filamentous fungi. However, it is known that FK506 exhibits antifungal activity against Aspergillus species. Therefore, we have added the following information to the manuscript according to your suggestion.
“For filamentous fungi, FK506 demonstrated inhibitory effects on the growth of Aspergillus fumitatus in vitro. This was further supported by experiments using a mouse infection model, where the FK506-treated group showed significant survival extension compared with that of the control group [7].” (Lines 33–37)
- These are no data of Table 1 in the manuscript.
Response: We apologize for the oversight. We have added Table 1 to the manuscript.
Other comments:
- Line 67: Thereference of the manufacturer’s instructions should be provided.
Response: The usage protocol for MitoSOX Red Mitochondrial Superoxide Indicator (Life Technologies) has been described in lines 82–85.
- Line 106: All the statistical were performed using which software?
Response: All analyses were conducted with R version 3.6.3. According to your suggestion, we have added this information as follows:
“Analyses were conducted using R version 3.6.3 (R Foundation for Statistical Computing, Vienna, Austria).” (Lines 123–124)
- Line 122, 141: The data should be shown with mean ± standard deviation.
Response: According to your suggestion, we have added the standard deviation.
“57.8 + 23.7% at 3 h and 42.5 + 14.4% at 6 h (Fig. 1b).” (Lines 141-124)
- Line 248-249: Reference should be provided.
Response: Following your suggestions, upon reviewing the discussion, we have decided to delete this section.
Reviewer 3 Report
The manuscript describes the antifungal effect of an immunosuppressant and calcineurin inhibitor, Tacrolimus (FK506) on Candida parapsilosis. They used various assays to establish that it is fungicidal to the Candida species by inducing apoptosis. The set of assays and experiments performed are well done and supported the results. However, overall presentation of the manuscript needs to be improved. I would suggest the authors add more background information in the introduction and results sections.
Comments
1. Authors need to reframe the abstract and introduction to make it more compelling in the sense that what was the objective and their main findings. The introduction needs to be more detailed.
2. I did not find table 1 in the manuscript.
3. It would be good to have the growth of these Candida species on the drug media plates.
4. Results sections are not elaborated. The headings are just the experimental design instead of the result of that section. Please give a heading which describes your outcome or conclusion where it is possible.
5. Please mention the number of cells and fields used for the viability assay in Figure 2.
6. Figure 2a- mention the dye name instead of the fluorescence in the figure panel.
7. Are you sure that scale bar is 100 um in Figure 4?
8. Figure 5 the images are too small to figure out anything. It would be good to have zoomed out images to show the fluorescence in few cells.
Some grammatical mistakes.
Author Response
Dear Reviewers,
Thank you very much for your suggestions and comments; we have modified the manuscript accordingly. The modified sentences have been highlighted in yellow in the manuscript.
Reviewer 3
The manuscript describes the antifungal effect of an immunosuppressant and calcineurin inhibitor, Tacrolimus (FK506) on Candida parapsilosis. They used various assays to establish that it is fungicidal to the Candida species by inducing apoptosis. The set of assays and experiments performed are well done and supported the results. However, overall presentation of the manuscript needs to be improved. I would suggest the authors add more background information in the introduction and results sections.
Comments
- Authors need to reframe the abstract and introduction to make it more compelling in the sense that what was the objective and their main findings. The introduction needs to be more detailed.
Response: The details in the Introduction section have been elaborated more. (Lines 33-54).
- I did not find table 1 in the manuscript.
Response: We apologize for the oversight. We have added Table 1 to the manuscript.
- It would be good to have the growth of these Candida species on the drug media plates.
Response: Thank you very much for your comments.
- Results sections are not elaborated. The headings are just the experimental design instead of the result of that section. Please give a heading which describes your outcome or conclusion where it is possible.
Response: Following your suggestions, we have added more text (Lines 33-37). However, since this paper follows the "Communication" format, we believe the current amount of text is appropriate.
- Please mention the number of cells and fields used for the viability assay in Figure 2.
Response: Approximately 300 cells were examined in this study. The sentence has been added. (Line 169)
- Figure 2a- mention the dye name instead of the fluorescence in the figure panel.
Response: MitoSOX cited in Fig. 2a is dye name.
- Are you sure that scale bar is 100 um in Figure 4?
Response: There was a typographical error. The correct value is 10 um, not 100 um. This has been corrected. (Line 199)
- Figure 5 the images are too small to figure out anything. It would be good to have zoomed out images to show the fluorescence in few cells.
Response: We have made the figures larger. This should now make it easier for the readers to view.